Machine learning aided multiscale modelling of the HIV-1 infection in the presence of NRTI therapy

Tunc Huseyin 1
Sari Murat 2
Kotil Seyfullah enesseyfullah.kotil@med.bau.edu.tr enesseyfullah.kotil@boun.edu.tr 3 4
1 Department of Biostatistics and Medical Informatics, School of Medicine, Bahcesehir University , Istanbul , Turkey
2 Mathematics Engineering, Faculty of Science and Letters, Istanbul Technical University , Istanbul , Turkey
3 Department of Biophysics, School of Medicine, Bahcesehir University , Istanbul , Turkey
4 Department of Molecular Biology and Genetics, Faculty of Arts and Sciences, Bogazici University , Istanbul , Turkey
Orlov Yuriy
Electronic publication date: 2023 Mar 31
Publication date: 2023
Volume: 11
Electronic Location ID: e15033
Received 2022 Jun 17; Accepted 2023 Feb 19
Copyright: ©2023 Tunc et al.
Copyright year: 2023
Copyright holder: Tunc et al.
License: This is an open access article distributed under the terms of the Creative Commons Attribution License, which permits unrestricted use, distribution, reproduction and adaptation in any medium and for any purpose provided that it is properly attributed. For attribution, the original author(s), title, publication source (PeerJ) and either DOI or URL of the article must be cited.
License URL: https://creativecommons.org/licenses/by/4.0/

Keywords: AIDS, HIV infection, Machine learning, NRTI therapy, Mathematical models

Funding: TUBITAK, 2232—International Fellowship for Outstanding Researchers, Project number 118C244 This work was supported by TUBITAK, 2232—International Fellowship for Outstanding Researchers, Project number 118C244. All the results are the sole responsibility of the authors. The funders had no role in study design, data collection and analysis, decision to publish, or preparation of the manuscript.

==============================
Human Immunodeficiency Virus (HIV) is one of the most common chronic infectious diseases in humans. Extending the expected lifetime of patients depends on the use of optimal antiretroviral therapies. Emergence of the drug-resistant strains can reduce the effectiveness of treatments and lead to Acquired Immunodeficiency Syndrome (AIDS), even with antiretroviral therapy. Investigating the genotype-phenotype relationship is a crucial process for optimizing the therapy protocols of the patients. Here, a mathematical modelling framework is proposed to address the impact of existing mutations, timing of initiation, and adherence levels of nucleotide reverse transcriptase inhibitors (NRTIs) on the evolutionary dynamics of the virus strains. For the first time, the existing Stanford HIV drug resistance data have been combined with a multi-strain within-host ordinary differential equation (ODE) model to track the dynamics of the most common NRTI-resistant strains. Overall, the D4T-3TC, D4T-AZT and TDF-D4T drug combinations have been shown to provide higher success rates in preventing treatment failure and further drug resistance. The results are in line with the genotype-phenotype data and pharmacokinetic parameters of the NRTI inhibitors. Moreover, we show that the undetectable mutant strains at the diagnosis have a significant effect on the success/failure rates of the NRTI treatments. Predictions on undetectable strains through our multi-strain within-host model yielded the possible role of viral evolution on the treatment outcomes. It has been recognized that the improvement of multi-scale models can contribute to the understanding of the evolutionary dynamics, and treatment options, and potentially increase the reliability of genotype-phenotype models.

Introduction

Antiretroviral drug resistance is one of the main barriers to therapy success for HIV-positive patients. According to the WHO, the HIV drug resistance report 2021, 10% and 40% of adults are affected by drug-resistant strains (DRS) for naive and treated patients, respectively. In addition, 50% of newly diagnosed infants were exposed to the DRS. The DRS can be acquired with nonadherence to the therapy protocols, or patients can directly be infected with DRSs (Blower et al., 2001). Both scenarios yield life-long persistence of the DRS and need to be carefully tracked.

Quantitative evaluation of HIV drug resistance has been carried out with the use of phenosense assays by finding the fold-change of IC50 values (the amount of concentration to inhibit 50% of virion) between drug-resistant and wild-type strains (Zhang et al., 2005; Pham et al., 2018; Feng et al., 2016). Data modelling frameworks have been used to construct general mathematical relations between genotype and phenotype information (Tarasova et al., 2018; Steiner, Gibson & Crandall, 2020; Shah et al., 2020; Tarasova et al., 2023; Lagunin et al., 2023). These mathematical models aim to generalize the given data by means of encoding the amino acid sequence of target enzymes (Rhee, Taylor & Fessel, 2010). One of the main contributions of the current study is to explore how these models can be embedded into a within-host model to simulate the evolutionary dynamics of HIV strains. In particular, we simulate thousands of clinically relevant HIV mutant strains provided by Rhee, Taylor & Fessel (2010).

For forecasting the viral dynamics of HIV, various within-host models have been presented in ordinary differential equation (ODE) forms in the presence/absence of resistant strains and antiretroviral therapy (Perelson & Nelson, 1999; Dixit & Perelson, 2004; Rong, Feng & Perelson, 2007; Hadjiandreou, Conejeros & Vassiliadis, 2007; Sutimin et al., 2017; Wu & Zhao, 2020; Chen, Teng & Zhang, 2021). Perelson & Nelson (1999) proposed HIV within-host models consisting of CD4+ T cells (T), infected CD4+ T cells (T∗), macrophage cells (M), infected macrophage cells (M∗) and virions (V) in the presence and absence of antiretroviral therapy. Dixit & Perelson (2004) proposed T—T∗—V model by considering time-dependent intracellular efficiency of reverse transcriptase and protease inhibitors (RTIs and PIs). Rong, Feng & Perelson (2007) derived two-strain extension of the within-host model given in the literature (Perelson & Nelson, 1999; Dixit & Perelson, 2004) with antiretroviral therapy. Hadjiandreou, Conejeros & Vassiliadis (2007) revised the T—T∗—M—M∗—V model proposed by Perelson & Nelson (1999) by adding homeostatic cell proliferation terms to capture long time behaviour of the HIV dynamics. Sutimin et al. (2017) modeled the within-host HIV dynamics with target Langerhans and CD4+ T cells and investigated the time-dependent efficiency of RTIs and PIs with various scenarios. Wu & Zhao (2020) derived two-strain within-host model including the age of infection detail represented by the system of integro-differential equations. They mathematically formulated the competition between the drug-sensitive and drug-resistance strains with respect to model parameters. Chen, Teng & Zhang (2021) included saturated incidence and distributed infection delays into the standard two-strain T—T∗—V model and investigated the effects of those novel incidence terms on the long-time behaviour of the dynamics. Additionally, the effect of drug adherence on the virological failure of ARTs (Rosenbloom et al., 2012), the effect of time-dependent drug efficiencies on ART response (Rong, Feng & Perelson, 2007; Vaidya & Rong, 2017), competition between susceptible and resistant strains in the viral dynamics (Ball, Gilchrist & Coombs, 2007; Lythgoe, Pellis & Fraser, 2013), the role of latently infected CD4+ T cell reservoirs on the evolution of strains (Doekes, Fraser & Lythgoe, 2017), comparison of entry inhibitors with the RTIs and PIs according to viral resistance (Alshorman, Al-hosainat & Jackson, 2022), investigation of optimal timing for ART Rouzine (2022) have been proposed through within-host models. The proposed mathematical models assume the co-existence of susceptible and resistant strains and generally investigate the response to antiretroviral therapy (ART). The current study addresses similar questions with a novel multiscale model based on Stanford HIV Drug Resistance data and machine learning models.

For the first time, we combined the experimental drug resistance data of nucleotide-reverse transcriptase inhibitors (NRTI) available in the Stanford HIV drug resistance database (Rhee et al., 2003) with a within-host model of HIV infection to observe the dynamics of the viral strains under different scenarios. Our multiscale model brings together three pieces of information: IC50 values for each mutant with machine learning models, within blood dynamics for NRTIs, and CD4+ T cells and macrophage cells for primary targets of virions. For different mutant compositions, we aim to investigate the emergence of treatment failure for different initiation timing (up to one year) and adherence level of NRTI therapies (21 different combinations). Here we rank the inhibitory capabilities of the NRTI combinations in the presence of various viral strains and ongoing viral evolution. Our results add to the predictions of the Stanford HIV drug resistance database, which identifies the best drug by selecting the one that has the lowest IC50 for a given mutant. But that model is a static model that cannot incorporate the effects of new mutants that can be generated through time which is accounted for in our model.

Materials and Methods

Within-host model with wild-type virus

In this part, we have inspired from the earlier studies on the within-host HIV infection model (Hadjiandreou, Conejeros & Vassiliadis, 2007; Hernandez-Vargas, 2019; Hernandez-Vargas & Middleton, 2013). We assume that the primary reservoirs for HIV infection are: CD4+T cells and macrophages denoted by T(t) and Mt (Hernandez-Vargas, 2019; Hernandez-Vargas & Middleton, 2013). The long-living macrophage cells cause the persistence of virions over the years (Orenstein, 2001; Herbein & Varin, 2010). Macrophage cells contribute to the depletion of healthy CD4 + T cells in advanced HIV infection (Crowe, 1995). Within-host modelling of HIV infection without considering the macrophage reservoirs yielded less reliable dynamics, such as the models that never result in the AIDS phase (Rong, Feng & Perelson, 2007). We denote the HIV infected CD4+ T cells and macrophages by T∗(t) and M∗t. Lastly, the number of free wild-type virions in the host is denoted by the function Vt. By considering model assumptions like homeostatic cell proliferation terms (sT,  sM), bilinear incidence terms (kTTV,  kMTM),  natural deaths of cells and virions (δTT,  δMM,  δT∗T∗,  δM∗M∗,  δVV), viral replication terms (pTT∗,  pMM∗) and the Michaelis–Menten type proliferation terms ρTVcT+VT,ρMVcM+VM, we express the one strain within-host model with the following system of ordinary differential equations (Hernandez-Vargas, 2019; Hernandez-Vargas & Middleton, 2013) dTdt=sT−kTTV−δTT+ρTVcT+VT

dT∗dt=kTTV−δT∗T∗

(1) dMdt=sM−kMMV−δMM+ρMVcM+VM

dM∗dt=kMMV−δM∗M∗

dVdt=pTT∗+pMM∗−δVV

where initial conditions are considered as T0=T0,T∗0=T0∗,M0=M0,M∗0=M0∗ and V0=V0. Further details of the model (1) can be seen in the study of Hernandez-Vargas & Middleton (2013). In the following section, we expand the model Eq. (1) to include both susceptible and resistant multiple strains as well as NRTI therapy.

Multi strain within-host model with NRTI therapy

The ARTs include at least one of the NRTIs that aim to block the activation of the reverse transcriptase enzyme. Effective treatment of HIV-positive patients with NRTIs saves millions of lives worldwide (Tressler & Godfrey, 2012). However, the error-prone structure of the HIV replication yields resistant strains over the years, and these strains are known to be a primary barrier to preventing AIDS (Kuritzkes, 2011). Our multiscale within-host model includes three main steps: constructing machine learning models to generalize isolate-fold change data for NRTIs, a model for dealing with NRTI action in blood, and finally, a within-host model with multi-strains and NRTI therapy.

An artificial neural network model for isolate-fold change relation

There exists various genotype-phenotype experiment data, including the fold change values of IC50 (the required drug concentration to inhibit 50% of virions) for various reverse transcriptase inhibitors in the presence of susceptible and resistant isolates (Rhee et al., 2005). The most used genotype-phenotype data is the Stanford HIV drug resistance database (https://hivdb.stanford.edu/). We use filtered genotype-phenotype data of reverse transcriptase inhibitors available in this database and are widely used for various machine learning algorithms (Amamuddy, Bishop & Bishop, 2017; Masso & Vaisman, 2013). By regulating the data for each NRTI, 1,224 unique mutations were observed for the reverse transcriptase enzyme. In this filtered dataset, 1,662 isolates for epivir (3TC), 1,597 isolates for abacavir (ABC), 1,683 isolates for zidovudine (AZT), 1,693 isolates for stavudin (D4T), 1,693 isolates for didanosine (DDI) and 1,354 isolates for tenofovir (TDF) have been analyzed for NRTI susceptibility. The dataset includes 1,206, 1,136, 1,220, 1,223, 1,223, and 1,119 unique mutations for 3TC, ABC, AZT, D4T, DDI, and TDF, respectively.

Here, we apply the binary barcoding technique (Rhee, Taylor & Fessel, 2010) to represent the isolates occurring in the dataset. Hence, 1,224-dimensional input vectors of 0s and 1s are created by considering the existence of unique mutations in the isolates. Let us denote our complete mutation set as M=m1,m2,…,m1224 where mi is an NRTI specified mutation pattern. We define the binary representation of isolate j as Ij=a1,a2,…,a1224 with ak=1,ifmk∈Ij0,otherwise.

We construct six artificial neural networks (ANN) models to predict logarithmic fold-change values in the presence of any isolates related to each NRTI therapy by using the Machine Learning and Deep Learning toolbox of the MATLAB 2022a program (https://www.mathworks.com/). The ANN architectures include 1,224-dimensional input, five hidden layer neurons, and one output neuron with hyperbolic tangent-sigmoid and linear activation functions. The model selection process is explained with detailed quantitative observations in Table S1. The scaled conjugate gradient algorithm with MATLAB built-in function “trainscg” has been used in the training process over GPU. Let us denote our model as a function that maps isolate vectors to the fold changes as FoldChange=ANNXI

where I ∈ {0, 1}1×1224 and X is a specified inhibitor (X ∈ {3TC, ABC, AZT, D4T, DDI, TDF}). To overcome possible overfitting, we have implemented an ensemble learning process. For each inhibitor, the 50 ×100 model has been trained with random training, validation, and test set (80%, 10% and 10%). A model is chosen from every 100 models that yield the minimum mean square error for the test set of the corresponding inhibitor data. Hence, 50 optimal models are selected out of 5,000 models for each NRTI inhibitor, and the final model is calculated as the average of these models.

The prediction performance of six ANNX(I) models with linear correlation coefficient (R) and mean square error (MSE) values are presented in Fig. 1. According to the figure, ANNX(I) models yield accurate predictions with high R and low MSE scores. Mean MSE value of ANNX(I) models have been obtained as 0.0453 with 95% CI [0.0005–0.0901]. Similarly, the mean R value of the models has been calculated as 0.9093 with 95% CI [0.8677–0.9509]. To observe how six ANNX(I) models classify resistant and susceptible strains, we convert our regression models into classification models by labeling the data as resistant (Fold Change ≥ 3) and susceptible (Fold Change < 3). The receiving operating curves (ROC) corresponding to the six ANN models and the area under the curve (AUC) values are presented in Fig. S1. According to the classification results, we get the mean AUC score as 0.9649 with 95% CI [0.9423–0.9875]. Additionally, to see why such a nonlinear model is needed to map the genotype data into the phenotype output, we also perform multiple linear regression (MLR) analysis (with 20% holdout data) for data of six NRTIs. The regression and classification performance of the MLR models are shown in Figs. S2–S3. A fair comparison between the ANN and MLR models in terms of the MSE, R, and AUC values is given in Table S2. According to the table, even classification performance of the models is almost the same, the ANN models give much more accurate estimations in regression. Since better regression performance is more desirable for our further modelling framework, the ANN models are assumed to be our baseline models for predicting the resistance profiles of given viral strains.

Figure 1 Regression performance of the six ANN models for each NRTI to predict logarithmic fold change values (log(FC)) of the mutant strains existing in the data.

The x-axis of the figures denotes logarithmic fold change value, which is mathematically equivalent to logIC50mutantIC50wild−type, for all existing mutant strains in the data and y-axis denotes corresponding predictions of the ANN models. For each ANN model, linear correlation coefficient (R) and mean square error (MSE) metrics are specified to measure the ability of these models to fit the existing real data.

Modelling the time-dependent drug efficacy

Modelling the efficacy of antiretrovirals using the plasma drug concentrations can be seen in various studies in the literature (Dixit & Perelson, 2004; Rong, Feng & Perelson, 2007; Rosenbloom et al., 2012). Rosenbloom et al. (2012) modeled the time-dependent drug efficiency in plasma by considering the exponential decay of plasma drug concentration after the instantaneous peak. Here we use the time-dependent drug efficacy model described by Dixit & Perelson (2004) and Rong, Feng & Perelson (2007), considering the pharmacokinetic parameters of drugs in the blood. Dixit & Perelson (2004) considered the phosphorylated concentration of the tenofovir (TDF) in the cells. Since the time-drug efficiency functions obtained by taking into account blood concentration and phosphorylated within cell concentration of drugs follow a very similar trend, here we assume the blood concentration of the drugs (see Fig. 1 of Dixit & Perelson (2004)). Additionally, the non-availability of phosphorylation reaction parameters for the remaining five inhibitors 3TC, ABC, AZT, D4T, and DDI have encouraged us to consider the blood concentration of the drugs only.

Let ɛXYt denotes the time-dependent efficacy of drug X in the presence of strain (isolate) Y. The instantaneous efficacy can be approximated as Dixit & Perelson (2004) (2) ɛXYt=CbXtIC50XY+CbXt

where CbXt denotes the within blood concentration of drug X and IC50XY denotes the required concentration of drug X to inhibit the 50% of strain Y. According to our isolate-fold change ANN model, Eq. (2) can be rewritten as (3) ɛXYt=CbXtANNXYIC50XWT+CbXt

where IC50XWT denotes the required concentration of drug X to inhibit the 50% wild type virus. Thus, to completely describe ɛXYt, we should model CbXt. According to Dixit & Perelson (2004), the concentration of a drug in the blood can be expressed as (4) Cbt=FDkae−ketVdke−kaekaId−11−eke−kat1−eNdkaId+ekeId−ekaIdeNd−1keId−1ekeId−1−eNd−1ke+kaId

where F is the bioavailability of the drug, D is the mass of the drug administered in one dose, Id is the dosing interval, Nd is the number of doses up to time t, Vd is the volume of distribution, ka and ke are pharmacokinetic parameters. The drug-specific parameters ka,  ke,  D,  Id and F occurred in Eq. (4) and IC50 values for 3TC, ABC, AZT, D4T, DDI, and TDF according to the equations given by Dixit & Perelson (2004) are evaluated and presented in Table 1. Detailed explanations of the derivation of these parameters are given in the Supplementary Information.

A multi-strain within-host model

This part of the study combines our investigations into a unique multi-strain within-host model. To reduce the cost of the simulations, we assume the main NRTI-related mutations 115F, 151M, 184I, 184V, 210W, 215F, 215Y, 41L, 65N, 65R, 67N, 69D, 70E, 70G, 70R, 74I and 74V according to the study of Rhee et al. (2005). These 17 mutations yield 131,071 unique strains having all possible mutations. Thus, by considering wild-type and mutant strains, we have total N = 131,072 strains. Our multi-strain within-host model with time-dependent NRTI therapy can be derived from one strain model (1) as follows dTdt=sT−kTT∑i=1N1−ci1−ɛXitVi−δTT+ρT ∑i=1NVicT+ ∑i=1NViT

dTi∗dt=kT1−ci1−ɛXitTVi−δT∗Ti∗

(5) dMdt=sM−kMM ∑i=1N1−ci1−ɛXitVi−δMM+ρM ∑i=1NVicM+ ∑i=1NViM

dMi∗dt=kM1−ci1−ɛXitMVi−δM∗M∗

dVidt=pTTi∗+pMMi∗−δVVi

Table 1 Drug specific parameters for time-dependent drug efficiency equation Eq. (4).

Parameter/Drug	3TC	ABC	AZT	D4T	DDI	TDF	
IC50(×10−5 mg/ml)	3.97	132.64	1.87	4.25	113.11	16.24	
D (mg)	300	300	300	40	400	300	
Id (day)	1	0.5	0.5	0.5	1	1	
F	0.86	0.83	0.64	0.86	0.42	0.39	
k a	27.98	51.07	37.42	54.29	32.34	8.36	
k e	3.44	8.52	14.25	7.84	47.30	16.58	
Vd (ml)	91,000	60,200	112,000	46,000	54,000	87,500	

where i = 1, 2, …, N =131,072, Ti∗t and Mi∗t denote the number of CD4 + T cells and macrophage cells infected by strain i and Vi(t) represents the number of virions having i th genotype. In the multi-strain within-host model (5), ɛXit denotes the time-dependent efficacy of the inhibitor X on the strain i and 0 ≤ ci ≤ 1 represents the fitness costs of mutant strains with c1 = 1 for the wild type of strain. The lack of enough experimental results on these fitness values compelled us to use the mean fitness cost values of mutations 41L, 67N, 70R, 184V, 210W, 215D, 215S, and 219Q estimated by Kühnert et al. (2018) as 0.2232, 0.3181, 0.3863, 0.5899, 0.3091, 0.0981, 0.1664 and 0.3207, respectively. According to these data, we assume that ci = 0.3015 for mutant strains i ≥ 2. A schematic illustration of the multi-strain within-host model (5) is given in Fig. 2. Parameter values of multi-strain within-host model (5) with corresponding references can be seen in Table 2.

Figure 2 Illustration of the core parts of multi-strain within-host model (5) with NRTI therapy.

Model (5) assumes the healthy CD4 + T cells Tt and macrophage cells Mt as the main targets of the viral strains Vit. Tt and Mt increase with both homeostatic cell proliferation and cell proliferation due to the increasing viral load. Viral strains infect both CD4 + T cells and macrophage cells and then those healthy cells become infected CD4 + T cells Ti∗t and macrophage cells Mi∗t. Ti∗t and Mi∗t compartments produce mature viral strains Vit with some constant rates. All compartments have natural death or clearance with some constant rates. NRTIs block the infection mechanism of the viral strains in healthy cells. The efficiency of the NRTIs is estimated through pharmacokinetic Eq. (3) and the pre-trained artificial neural network models that map the genotype data to fold-change values of the IC50’s with respect to the wild type virion.

The within-host model (5) ignores the role of latently infected CD4+ T cells. The main role of latently infected CD4+ T cells is the viral rebound after poor adherence to the given therapy (Chun et al., 2000), and these cells are almost three percent of all CD4+ T cells (Hadjiandreou, Conejeros & Wilson, 2009). Since model (5) is continuous over time and hence the emerged viral strains are not completely eradicated in the viral suppression phase, the persistence of HIV-1 virions is automatically ensured, and poor adherence in model (5) provides viral rebound. Thus, ignoring the latently infected CD4+ T cells in model (5) does not considerably affect our modelling framework. As indicated in the study of Chun et al. (2000), latently infected CD4+ T cells are not the only reason for the rebound of plasma viremia after discontinuation of the ART. The literature (Alexaki, Liu & Wigdahl, 2008; Hendricks et al., 2021; Kruize & Kootstra, 2019) show that the macrophage cells are of particular importance in HIV-1 persistence, and this is why model (5) considers this observation like some existing studies (Hadjiandreou, Conejeros & Vassiliadis, 2007; Hadjiandreou, Conejeros & Wilson, 2009; Hernandez-Vargas, 2019; Hernandez-Vargas & Middleton, 2013). Consideration of the role of the macrophage cells yields slow progression to the AIDS phase for untreated patients and improves the reliability of model outcomes (Hernandez-Vargas, 2019).

Table 2 Parameter values, units and parameter intervals of the within-host models and taken from the literature (Hernandez-Vargas & Middleton, 2013; Hernandez-Vargas, 2019).

Parameter	Value	Unit	Parameter variation	
s T	104	ml −1 d −1	7 × 103 − 2 × 104	
s M	150	ml −1 d −1	100 − 300	
k T	4.5714 × 10−8	mld −1	3.2 × 10−8 − 10−7	
k M	4.3333 × 10−11	mld −1	1.73 × 10−11 − 1.3 × 10−9	
p T	38	d −1	30.4 − 114	
p M	35	d −1	22 − 132	
δ T	0.01	d −1	0.001 − 0.017	
δ T ∗	0.4	d −1	0.1 − 0.45	
δ M	0.001	d −1	10−4 − 1.4 × 10−3	
δ M ∗	0.001	d −1	10−4 − 1.2 × 10−3	
δ V	2.4	d −1	0.96 − 2.64	
ρ T	0.01	d −1	–	
ρ M	0.003	d −1	–	
c T	3 × 105	ml −1	–	
c M	2.2 × 105	ml −1	-	

To model the effect of mutations, we do not explicitly include the mutation matrix in the ODE system (5); instead, we address the transition between mutations and strains at the end of each time step by generating Poisson random numbers (Rosenbloom et al., 2012). Let us assume time step n (t = n day), Ti∗n=Ti∗n and Mi∗n=Mi∗n. The mutation matrix of our system is denoted by MT and defined as (6) MTij=1,ifstrainicantakeamutationtobecomestrainj0,otherwise

For the infected CD4 T cells Ti∗n and infected macrophage cells Mi∗n, we calculate the number of new infected ones in one day period as ΔTi∗n and ΔMi∗n without taking into account the death of these newly infected cells. For each i = 1, 2, …, N, poissrndμΔTi∗n and poissrndμΔTi∗n number of infected cells are randomly transmitted from strain i to strain j according to the mutation matrix MTij where function poissrnd(x) generates Poisson random number with mean x and μ = 3 × 10−5 denoting the mutation rate (Rosenbloom et al., 2012). Note that the mutation rate for each point mutation is unique for the corresponding amino acid change, but we assume a fixed average mutation rate μ = 3 × 10−5 as stated by Rosenbloom et al. (2012). Since NRTI-related mutation rates have low variance value (Rosenbloom et al., 2012) and we have so many viral strains to track, we use overall mutation rate μ = 3 × 10−5. Parameter values of models (1) and (5) are presented with their references in Table 2.

Model (5) can also include dual therapy of NRTIs X and Y by modifying the therapy-related time-dependent infection coefficients for CD4 + T cells and macrophage cells βiT/Mt=kT/M1−ci1−ɛXit with the use of Bliss independence of drug actions as Jilek et al. (2012) (7) βiT/MɛXit,ɛYit=kT/M1−ci1−ɛXit1−ɛYit

or Loewe additivity of drug actions (Jilek et al., 2012) (8) βiT/MɛXit,ɛYit=kT/M1−ci1ɛXit1−ɛXit+ɛYit1−ɛYit+1.

Bliss independence assumes independent actions of combined drugs, and Loewe additivity assumes the competition for the same binding site. According to Jilek et al. (2012), all combinations except AZT-D4T and DDI-TDF obey the Bliss independence rule, and these two combinations obey the Loewe additivity rule. Note that, since we assume kM ≈ kT/1000 and βiTt≈βiMt/1000 according to the Hernandez-Vargas (2019) and Hernandez-Vargas & Middleton (2013) (see Table 2), we prefer to use the notation βi for βiT throughout the following parts. Whenever βi values are quantitatively mentioned in the results section, these values correspond to the βiT.

Note that even though we describe our model parameters for 1 ml of blood in Table 2 as widely assumed in the literature (Hadjiandreou, Conejeros & Vassiliadis, 2007; Hernandez-Vargas, 2019; Hernandez-Vargas & Middleton, 2013), we simulate the viral dynamics in the host plasma (3,000 ml; Rosenbloom et al., 2012) to catch more viral diversity. We assume that the only reservoir of HIV virions is the plasma, which is the major one (Valcour et al., 2012), even if there exist other reservoirs like lymph nodes or cerebrospinal fluid (CSF) (Valcour et al., 2012; Haase, 1999). Since the instantaneous drug efficiency rates are (ɛXYt) in non-dimensionless form, we can easily simulate the dynamics in the host plasma by converting the volume-dependent model parameters given in Table 2. For example, by considering 3L host plasma (Rosenbloom et al., 2012), the infectivity parameter kT = 4.5714 × 10−8 ml/day equivalently becomes kT=4.5714×10−83000 plasma/day = 1.5238 × 10−11 plasma/day.

Results

This section provides the simulation results of the multi-strain within-host model (5), starting with various viral strains. The effects of adherence levels and initiation timing of NRTI therapies on the progression of viral dynamics are investigated. This section includes four subsections in which we propose the statistics of the infection rates, details of model simulations, the quantitative measure for the therapy success, and the simulation results for various cases.

Statistics of infection rates

Before running the simulations to observe the failure/success distribution of each NRTI combination, we may predict the best possible therapy protocol through our pre-trained machine learning model and the pharmacokinetic properties of the inhibitors. Obviously, as we infer from our model (5) and drug-specific time-dependent infection rate βiɛXit,ɛYit (7)–(8), each viral strain has its infection rate and aims to be dominant by infecting more healthy cells. Since evaluation of βiɛXit,ɛYit is straightforward through Eqs. (7)–(8) and (3), we may have some prior estimates for the selection of the best therapy protocol. Distribution of 131,071 βiɛXi,ɛYi= ∫01βiɛXit,ɛYitdt values in the presence of 21 different mono and dual NRTI therapies are illustrated in Fig. 3. Descriptive statistic values of βiɛXi,ɛYi values for all combinations are presented in Table 3.

Figure 3 Probability distributions of infection rate (βi) values of various viral strains in the presence of NRTI therapy combinations.

(βi) values are calculated with Eqs. (7)–(8) depending on the drug pairs. (βi) values are effected by pharmacokinetic parameters, IC50 values for the viral strains, baseline infection rate kT = 4.5714 × 108 and the fixed viral fitness value (ci = 0.3015) of the viral strains.

Table 3 Descriptive statistics (×10−8) of infection rate βi values for all possible mono and dual NRTI therapies.

Drugs	Mean	Min	Max	Std	Median	Mode	Q 1	Q 3	
D4T-3TC	1.290	0.160	2.069	0.363	1.295	0.16	1.029	1.576	
D4T-AZT	1.370	0.427	2.358	0.442	1.368	0.427	1.021	1.722	
TDF-D4T	1.403	0.604	2.373	0.371	1.382	0.604	1.109	1.679	
D4T	1.442	0.697	2.319	0.351	1.426	0.697	1.157	1.72	
AZT-3TC	1.473	0.154	2.212	0.462	1.531	0.154	1.109	1.877	
D4T-ABC	1.525	0.695	2.405	0.374	1.514	0.695	1.221	1.826	
DDI-D4T	1.592	0.765	2.466	0.382	1.581	0.765	1.279	1.903	
TDF-AZT	1.627	0.474	2.523	0.506	1.683	0.474	1.226	2.056	
AZT-ABC	1.755	0.551	2.529	0.503	1.844	0.551	1.363	2.194	
AZT	1.775	0.554	2.564	0.513	1.858	0.554	1.373	2.223	
DDI-AZT	1.834	0.562	2.602	0.533	1.933	0.562	1.412	2.307	
TDF-3TC	1.884	0.3	2.265	0.307	1.952	0.3	1.835	2.065	
3TC	1.965	0.29	2.173	0.339	2.114	0.29	2.000	2.133	
ABC-3TC	2.030	0.274	2.318	0.359	2.173	0.274	2.025	2.225	
DDI-3TC	2.155	0.323	2.373	0.356	2.305	0.323	2.201	2.325	
TDF-ABC	2.172	1.508	2.588	0.181	2.176	1.508	2.038	2.314	
TDF	2.299	1.889	2.665	0.172	2.3	1.889	2.163	2.438	
TDF-DDI	2.323	1.917	2.667	0.164	2.327	1.917	2.194	2.457	
ABC	2.459	1.733	2.698	0.126	2.485	1.733	2.404	2.544	
DDI-ABC	2.546	1.869	2.726	0.106	2.57	1.869	2.505	2.617	
DDI	2.746	2.675	2.78	0.02	2.747	2.675	2.732	2.762	

Figure 3 and Table 3 show that the probability distributions are almost uniform and βiɛXi,ɛYi values have considerable diversity and standard deviations among the viral strains. Hence, this observation means that even having point mutations can change the infection rates considerably and thus may lead to a need for more perfect adherence levels to the given therapy. Additionally, Fig. 3 implies that the initial viral strain of the patient plays a critical role in the progression of HIV dynamics. According to Table 3, NRTI therapy combinations yield 38.4% and 78% decrease in infection rate on average (among all therapies) (95% CI [36.2%–40.7%] and [69.7%–86.3%]) for the worst and best case scenario (having most and least resistant initial strain), respectively.

Table 3 ranks the possible NRTI combinations in terms of the resistance scores but ignores the side effects and cost-effectiveness. Various side-effects of NRTIs linked with mitochondrial toxicity (Holec et al., 2018). We present the possible side-effects of the existing NRTIs in Table S3, and a detailed review can be found in the study of Montessori et al. (2004). The cost-effectiveness of NRTI therapies is essential to maximize the expected survival times of the patients with minimized costs. Various mathematical models are available that compare treatments for cost-effectiveness, and a detailed review of Mauskopf (2013) provides various essential results. Most of the models described in their study ignore the effect of drug resistance. Drug resistance is a crucial contributor to the expected costs. This study is only interested in the impact of drug resistance on the NRTI therapy outcomes, and we both ignore side effects and cost-effectiveness.

Details of model simulations

In our simulations, we investigate the effect of the type of NRTI therapy, timing of the NRTI therapy, and adherence to the provided therapy on CD4+ T cell counts of the patients. All possible 21 mono and dual NRTI combinations of six inhibitors have been included in the simulations by considering their independent or additive actions. The initiation time of the NRTI therapy is considered within the first year after the patient became infected and denoted by τ. The adherence level of a patient to the provided therapy protocol is assigned to a real number α between 0 and 1, representing nonadherence to full adherence levels. After initiating the treatment with adherence level α in a day of the simulation, the patient takes drug(s) with probability α according to the parameters given in Table 1. Initial viral load, CD4 + T cell count, and macrophage cell count in the simulations are considered as 1 virion/ml, 106 cell/ml and 150 cell/ml, respectively (Hernandez-Vargas, 2019).

It is assumed that the patient is infected with one type of mutant strain with one to five-point mutations on the reverse transcriptase enzymes. In this way, five groups are constructed to include five different strains. These viral strains have been determined according to the frequency of presence in the Stanford HIV drug resistance database. These initial viral strains are denoted by Gij where i = 1, 2, 3, 4, 5 denotes the number of the point mutations in the strain and j = 1, 2, 3, 4, 5 indexes the most frequently occurring examples in the dataset. We have performed our simulations with these 25 different initial viral strains having the following point mutations: G11=69D,G12=70E,G13=74I,  G14=151M,  G15=41L,  G21=69D,115F,G22=69D,215Y,  G23=70R,215Y,  G24=67N,69D,G25=67N,70R, G31=69D,115F,215Y, G32=69D,70R,115F, G33=67N,69D,215Y, G34=67N,70R,215Y, G35=67N,69D,70R, G41=67N,69D,115F,215Y, G42=67N,70R,115F,215Y,   G43=69D,70R,115F,215Y, G44=67N,69D,70R,115F, G45=65N,69D,70R,215Y, G51 = {65N},    {69D,  70R, 115F,  215Y}, G52=69D,70R,74F,115F,215Y, G53=41L,67N,69D,70R,215Y, G54=65N,67N,69D,70R,215Y, G55=67N,69D,70R,74I,215Y. For instance, G14=151M strain has only one point mutation 151M and the rest of the amino acids are the same as wild type HIV-1 virus.

Measuring the therapy success

It is essential to track the success of the given antiretroviral therapy by hindering the viral dynamics from the AIDS phase, i.e., by keeping the CD4 + T cell count as high as possible. The AIDS phase occurs when CD4 + T cell count is less than 200 cell/µl (Kitahata et al., 2009). Our primary criterion for the success of NRTI therapy is the occurrence and nonoccurrence of the AIDS phase after initiation of the therapy with some initiation timing τ and adherence level α, as was done in cohort studies (van Sighem et al., 2003). Note that it is also possible to increase the CD4 + T cell counts of patients during the AIDS phase by the initiation of the ARTs (Shoko & Chikobvu, 2019). However, here we are not analyzing what happens after the AIDS phase. Our primary goal is to determine how evolutionary dynamics under the NRTI therapy affect the occurrence of the AIDS phase.

All simulations start with one infected CD4 + T cell and one infected macrophage cell with one of the initial strains Gij. The simulation final time tf is considered 20 years, and therapy success/failure is determined according to the occurrence of the AIDS phase in 20 years. However, we note that the clinical goal of ART therapy is the full suppression of detectable viremia. In our simulations, total suppression of detectable viremia is equivalent to not developing AIDS after 20 years. However, the opposite is false: detectable (>200 copies/ml) suppression misses low copies of violent mutants, eventually leading to the AIDS phase. Therefore, we consider the AIDS occurrence as our output. In the clinic, therapy is redesigned if complete suppression is not observed. However, our simulations never redesign the treatment to distinguish between successful/failed drug combinations.

We run our simulations for randomly scattered 512 α,τ∈0,1×0,365 pairs for predetermined initial strain Gij. The success rate (SR) of a therapy is measured as the number of α,τ pairs that lead to protection from the AIDS phase in all 512 α,τ pairs. In Fig. 4, we show some representative simulation results of the multi-strain within-host model (5), starting with the G51=65N,69D,70R,115F,215Y strain under various mono and dual NRTI therapies with randomly scattered α,τ pairs. For this simulation setup, nine out of 21 NRTI therapy protocols have considerable success in preventing the patient from the AIDS phase. The importance of adherence level (α) and initiation timing (τ) is evident from the figure for all cases. In some cases, such as the DDI-D4T combination shown in Fig. 4, the initiation timing considerably affects the success rates. Higher τ values yield therapy failure even at high adherence levels. As observed from the figure, the D4T-3TC combination yields the best SR value by performing well for late initiation with perfect adherence levels. For the current case, the success of the D4T-3TC combination is mainly due to the behaviour of the therapy in the higher initiation timing (τ) region.

Figure 4 Illustration of possible mono and dual NRTI therapy outcomes carried out using 512 random α,τ pairs in the current multi-strain within-host model (5).

The initial strain has been selected as G51=65N,69D,70R,115F,215Y. Blue circles represent the failure after 20 years of simulation, i.e., the AIDS phase occurs when the patients start the therapy τ after infection and take the therapy with an adherence rate α. Purple squares mean that the therapy succeeds under the conditions mentioned above. SR values represent the success rate defined as SR = # of purple squares/# of all data points.

While the importance of the adherence levels is evident from its direct relation with infection rates, the importance of the initiation timing is non-evident and should be explained here clearly. In Fig. 5, we illustrate the effect of initiation timing τ in our multi-strain model (5) when initial strain and adherence level are selected as G51=65N,69D,70R,115F,215Y and α = 0.5. According to Figs. 5A–5B, τ = 50 yields successful therapy by maintaining healthy CD4 +T cell and macrophage cells at normal levels and declining the viral load to undetectable levels. On the other hand, when we assume the initiation timing as τ = 360, virologic failure and AIDS phase are observed in Figs. 5C–5D. According to our model (5), the main difference between early and late initiation timing is the diversity of viral strains at the initiation to therapy times. Late initiation to the therapy increases the probability of the occurrence of the more resistant strains, even if their ancestors are slowly growing. For example, as we compare Fig. 5B with Fig. 5D, the two generations of mutant strains occur when τ = 360 (Fig. 5D) while there exists only one generation of mutant strains when τ = 50 (Fig. 5B). The two generations of mutant strains yield viral rebound and failure of the therapy in Fig. 5D.

Figure 5 The effect of initiation timing is illustrated with healthy cell and virion counts.

The initial strain is taken as G51=65N,69D,70R,115F,215Y and the common adherence level α = 0.5 is considered. (A) Dynamics of T(t) and M(t) when τ = 50, (B) dynamics of viral strains when τ = 50,  (C) dynamics of T(t) and M(t) when τ = 360, (D) dynamics of viral strains when τ = 360. Black dashed vertical lines in parts c and d denote the HIV detection limit in blood as 200 copies/ml (Barletta, Edelman & Constantine, 2004).

If we go back to Fig. 4, the NRTI combinations having boundary lines with relatively low slope values are more sensitive to increasing values of τ since these therapies yield high variance in IC50 values of possible viral strains mutated from the initial strain. Therefore, in our modelling framework, the late initiation is directly related to the variance of IC50 values corresponding to the initial strain and possible mutants. Thus, the level and type of the NRTI therapy should be planned so that the reoccurrence of the viral strains should be blocked depending on the initiation time τ. Additionally, in the reoccurrence phase of viral strains, non-perfect adherence to the therapy leads to the selection of resistant strains (Fig. 5D). In this case, two possible problems arise:

1. If the therapy protocol of the patient is updated, therapy is less likely to be successful than when therapy was first started.

2. The probability of infecting another person with more resistant strains increases, and the probability of having an AIDS phase increases for the infected person.

The existence of low viral loads of new mutated strains is enough for selecting these strains after antiretroviral therapy. Therefore, according to our simulations, initiation timing is as crucial as the adherence level to overcome the AIDS phase and to protect the possible susceptible persons from more dangerous scenarios.

The NRTI mutants are known to have epistasis effects, which implies that the viral fitness of the mutant strain depends on the existing genetic background. The epistasis effects may lead to the selection of diverse branches in mutant generations (Biswas et al., 2019). Epistasis of mutations can impact the values of IC50 and fitness costs. The data we used to train our IC50 values implicitly includes epistatic effects. The ANN model that predicts IC50 values for mutants is expected to learn the epistatic interactions. However, it is not completely unlikely that some unobserved data may have unpredictable epistasis. Nevertheless, that variant being underrepresented in the data implies its irrelevance in the clinic. On the other hand, the fitness costs of mutants are assumed to be fixed due to a lack of enough data. Nevertheless, as we explain later, this assumption should not significantly impact our claims.

Simulation results

Here we have simulated our multi-strain within-host model (5) for all possible initial strains Gij to observe the effect of initial strains on success rates. All possible mono and dual NRTI therapies have been implemented for randomly scattered 512 α,τ∈0,1×0,365 pairs. The SR values of mono and dual NRTI therapies are calculated, and the well-performed combination results are comparatively illustrated in Fig. 6.

In line with Fig. 3 and Table 3, the D4T-3TC combination has been the best option for 20 out of 25 cases. The inhibitory potential of this combination is because of the pharmacokinetic parameters (see Table 1) of inhibitors, the drug-resistance profiles of inhibitors (see Table 3), and their Bliss-independent action on the target enzyme. Following the D4T-3TC combination, the TDF-D4T and D4T-AZT combinations are observed to be in first place in four and one out of 25 cases, respectively. The strong relation between the infection rate of an initial strain (and possible new strains) and the corresponding success rate value is evident from the correlation between Figs. 3 and 6. For instance, according to Fig. 3, the D4T-3TC combination yields fewer infection rates for most of the viral strains. Similarly, Fig. 6 shows that the D4T-3TC combination has great success rates for most of the initial viral strains. We will later quantitatively analyze the relationship between the infection rates of the detected viral strains and the success rates of the given therapies.

According to our modelling framework, since the fitness cost of all strains is assumed to be the same, the initial strain is dominant when the patient is diagnosed. Moreover, as evident from Figs. 5B–5D, considerable mutational variations at low copy numbers exist besides the initial strain. However, only the dominant strain is likely to be detected (strains having higher than 200 copies/ml in blood Barletta, Edelman & Constantine, 2004) when a phenosense assay is implemented. Thus, the clinician would only observe the initial strain and maybe a few mutational variations (according to Figs. 5B–5D, only the initial strain can be observed when the patient is diagnosed) to decide on the NRTI therapy protocol. Therefore, it is inevitable to ask whether the only predictor of the success rate is the detected viral strains at the diagnosis.

Figure 6 SR values of various NRTI combinations obtained by simulating multi-strain within-host model (5) with initial viral strain Gij for randomly scattered 512 α,τ∈0,1×0,365 pairs.

The undetected viral strains play a vital role in estimating the success rate and finding an optimal therapy protocol—especially their infection rates. We have trained regression models that predict therapy outcomes based on the infection rates of the initial strain and its mutants—the mutants will be referred to as first, second, third, fourth, and fifth generations. The first generation is mutated from the initial strain, whereas the second is mutated from the first. For the regression model, we aimed to determine how many generations of the detected strain(s) should be considered to predict an optimal therapy. To answer this question quantitatively, we construct the ANN and MLR models for predicting the success rate of therapy from the infection rates of the existing mutant strains. We construct six ANN and MLR models denoted by Gi for i = 0, 1, …, 5.Gi denotes i − th generation of the detected strain(s) that has been considered in the inputs of the models. For instance, model G0 only assumes the infection rates of the detected viral strain(s), and model G3 considers the infection rates of the detected viral strain(s) and the first three-generation mutants of this strain(s). In each generation of mutant strains, we use two values: mean and maximum values of the infection rates of the considered generation. Thus, together with the detected viral strain, the model Gi has 2i + 1 dimensional input. 2i input values denote the mean and maximum infection rates of i − th generation, and the remaining input value denotes the infection rate of the detected viral strain at the diagnosis. The graphical illustration of model Gi can be seen in Fig. 7.

Figure 7 Prediction process of SR values from the infection rates of the detected and possible mutant strains.

The models Gi are constructed by considering i generation of mutant strains and the detected strain itself. For each generation, mean and maximum values of the infection rates are assigned to the input of possible ANN and MLR models. SRANN and SRMLR denote the SR prediction of the ANN and MLR models from the given infection rate input.

Simulation results are given in Fig. 6 for 25 initial strains converted to the training data for the ANN and MLR models. 304 input–output relations have been obtained from various therapies having SR ≥ 0.02. For the ANN models, this data is divided into the train, test, and validation sets (70%, 15%, and 15%). Each Gi model having the ANN architecture is trained using the scaled conjugate gradient algorithm. Similarly, for the MLR models, 20% of the data is considered as a test set, and the remaining 80% is used in the training process. To test the prediction performances of the ANN and MLR models, we have generated an external test dataset by simulating the model (5) with 25 random initial strains having one-to-five-point mutations, and 314 test sample is obtained. Additionally, to observe how well our ANN and MLR models classify the therapies as successful (SR >  = 0.5) and unsuccessful (SR < 0.5), the area under the receiving operating curves is measured for both the ANN and MLR models.

We illustrate the regression and classification performances of the ANN models on the training and test sets in Fig. 8. Figure 9 shows similar predictive performance metrics of the MLR models on both the training and test sets. The mean square error (MSE), linear correlation coefficient (R), and area under the curve (AUC) metrics are presented for six Gi models having the ANN and MLR architectures. According to the test set performance of the models, model G2 gives better MSE, R, and AUC values with both the ANN and MLR architectures. That means considering the infection rates of both the detected strains and the first two mutant generations of the detected strains led to better predictions.

Figure 8 Regression and classification performances of models Gi having the ANN architectures on predicting the SR values of the therapies.

Models Gi assume the infection rates of the detected strain and its first i mutant generations and have 2i + 1 input values. Mean square error (MSE), linear correlation coefficient (R), and area under the curve (AUC) metrics are presented for both training and test data.

Figure 9 Regression and classification performances of models Gi having the MLR architectures on predicting the SR values of the therapies.

Models Gi assume the infection rates of the detected strain and its first i mutant generations and have 2i + 1 input values. Mean square error (MSE), linear correlation coefficient (R), and area under the curve (AUC) metrics are presented for both training and test data.

On the other hand, the G0 type models yield relatively poor regression and classification performances, i.e., considering only the infection rate of the detected strains is not enough to estimate better therapy protocols. This implies that the possible undetected mutant generations should also be taken into account in determining the therapy protocols. Nevertheless, there is a threshold on the number of mutant generations that must be considered. Figure 8 (for ANN architectures) and Fig. 9 (for MLR architectures) show that models G3, G4 and G5 overfit the data and yield less accurate predictions than the model G2 for both architectures. Additionally, for each Gi model, the ANN architecture yields a better approximation for the SR values than the MLR architectures.

Discussions and Conclusions

In this study, we have proposed a multi-strain within-host model of HIV infection with time-dependent NRTI therapy. Drug-resistant strains have been assumed to initiate the infection for the patients, and six available NRTI inhibitors with mono and dual combinations have been implemented in the simulations for various initiation timing and adherence levels. To assess the drug response curves with the IC50 values of the NRTI-resistant strains, artificial neural network models are trained for each inhibitor by using the Stanford HIV drug resistance database. To describe time-drug efficiency and time-infection rate curves, pharmacokinetic parameters of the inhibitors have been calculated and hybridized with the corresponding IC50 values. We have designed our simulation environment to determine the effect of initial strains, initiation timing for the therapy protocol, and adherence levels to the given drug usage schedule on the occurrence of the AIDS phase within 20 years after infection.

According to our modelling framework, the success rate of the NRTI therapies in case of late initiation has led to the availability of more resistant viral strains, and then the resistant strains become dominant in the host plasma after an initial decline of the detected strain. Although some mathematical models assume implicitly that the initiation timing does not affect the success-failure of the therapy (Dixit & Perelson, 2004; Rong, Feng & Perelson, 2007), our multi-strain model catches the penalty of late initiation since the late initiation was proven to block the therapy success in various experimental results (Kitahata et al., 2009; van Sighem et al., 2003). Our simulation results have shown that in the case of the late initiation to therapy, the efficiency of the therapy should be far more than the early initiation case to prevent the possible AIDS phase.

We have shown that D4T-3TC, D4T-AZT, and TDF-D4T combinations are less likely to result in treatment failure. These inhibitors have been seen to provide fewer infection rates due to their pharmacokinetic parameters and IC50 values in the presence of various viral strains. According to our results, the success rate of accurately predicting the best therapy depended on the composition of detected strains and their possible further mutants. This observation implies that the emergence of new mutants from the initial strain is likely to have a considerable effect on the success of the therapy. Thus, it is more reasonable to suggest the optimal therapy combinations for the patients by considering the detected viral strain and the undetected mutant, which most likely were generated from the detected strain.

The most important message of this article is that the undetected viral strains, at the diagnosis, may have considerable effects on therapy outcomes. Specifically, double mutants of the detected viral strain should be taken into account even if they were not detected. Earlier studies, such as Stanford HIVdb (Talbot et al., 2010), HIV-grade (Obermeier et al., 2012), REGA (Van Laethem et al., 2002), and ANRS (Agence Nationale de Recherches sur le SIDA, Meynard et al. (2002)) predicted the best possible therapy protocol. REGA is a rule-based model and was developed by scientists at Rega Institute for Medical Research and University Hospitals, and classifies the isolates as susceptible, intermediate, and resistant (Van Laethem et al., 2002). ANRS Meynard et al. (2002); Singh (2017) is also a rule-based computational resistance classifier based on a linear combination of mutations. However, the undetected viral strains may lower the prediction power of such models. We have shown that a multi-strain within-host model (5) can help estimate undetected mutant strains and their role in optimal therapy selection.

A possible criticism of our model is that each mutant strain should have a unique fitness cost. However, we assume a constant factor for all mutants. To our best knowledge, there is not much data for specific strains to construct a machine-learning model as we did for the IC50 values. According to the theory, fitness costs can play a role in selecting resistant strains, which can alter our success rate. However, the fitness costs would affect the dynamics more at low drug concentrations. Luckily, the phase changes (AIDS or no AIDS) occur at relatively high adherence levels, which implies a relatively high concentration.

Our modeled treatments include only NRTIs, but current clinical practice includes additional drugs (Aguilar et al., 2022). Indeed, including the other components of ART would add to the realism. However, it is known that different classes of HIV drugs generally interact independently (Rosenbloom et al., 2012; Jilek et al., 2012). By the independence assumption, the relative ranking of NRTI therapies is relevant to consideration for ART. However, we would like to openly indicate that our model is not designed to suggest a better first line of treatment but rather to relatively rank NRTI combinations in a multiscale model.

This study has investigated the effect of NRTI inhibitors, which are the most important members of Highly Active Antiretroviral Therapy (HAART) (Achhra & Boyd, 2013). Since the Stanford drug resistance database also includes the genotype-phenotype data of protease inhibitors (PI), non-nucleotide reverse transcriptase inhibitors (NNRTI), and integrase inhibitors (II), some future studies may include these groups of inhibitors with possible mono, dual or triple drug combinations. Some existing HAART protocols may also be simulated through such a modelling framework. On the other hand, we have not considered the too-late initiation of the NRTI therapy at considerably low CD4 + T cell levels because of the failure of simulated therapy protocols in such situations. Some future works may also investigate more comprehensive therapies to prevent patients from the AIDS phase when they are diagnosed too late.

Supplemental Information

Supplemental Information 1 Derivation of drug-specific parameters

Click here for additional data file.

Supplemental Information 2 The ANN hyperparameter tuning results

Mean square error (MSE) and linear correlation coefficient (R) for artificial neural network (ANN) models with different topologies (number of nodes per layer) for predicting logarithmic fold-change of IC50 values for the six NRTIs. Genotype-phenotype data of each NRTI is split as external test set and internal train-validation-test sets with 15% and 85%. Internal data sets are divided into training, validation, and test sets (80%, 10%, and %10). Internal data set is used for training and model selection. In the model selection process, we trained 20 models four times, and every 20 models have been tested on the internal test sets, and the models giving the best MSE scores are selected for each NRTI. Thus, four models are averaged to get the final model for each inhibitor. The selected models are tested on the external test set, and the resulting performance metrics are presented below. According to the table, two topologies yield relatively better results: the model has one layer with five neurons, and the model has two layers and three neurons in each layer. To make the model under consideration simple enough, we chose the model with a five-neuron hidden layer.

Click here for additional data file.

Supplemental Information 3 Comparison of the ANN and MLR models

Mean square error (MSE), linear correlation coefficient (R) and area under the curve (AUC) metric values for linear regression (LR) and artificial neural network (ANN) models for predicting fold-change values in IC50 values of the six NRTIs.

Click here for additional data file.

Supplemental Information 4 Pharmacokinetic parameters of the nucleotide reverse transcriptase inhibitors with considerable side effects

Click here for additional data file.

Supplemental Information 5 ROCs and corresponding AUC values for the ANN models

The ANN models are used to classify the given strains as resistant (Fold Change ≥ 3) and susceptible (Fold Change<3). The corresponding receiver operating characteristic curves with area under the curve (AUC) values are presented for each NRTI.

Click here for additional data file.

Supplemental Information 6 Regression performance of the six MLR models for each NRTI to predict logarithmic fold change values (log(FC)) of the mutant strains existing in the data

The x-axis of the figures denotes logarithmic fold change value for all existing mutant strains in the data and y-axis denotes corresponding predictions of the MLR models. For each MLR model, linear correlation coefficient (R) and mean square error (MSE) metrics are specified to measure the ability of these models to fit the existing real data.

Click here for additional data file.

Supplemental Information 7 ROCs and corresponding AUC values for the MLR models

The MLR models are used to classify the given strains as resistant (Fold Change ≥ 3) and susceptible (Fold Change<3). The corresponding receiver operating characteristic curves with area under the curve (AUC) values are presented for each NRTI.

Click here for additional data file.

Additional Information and Declarations

Competing Interests

Author Contributions

Data Availability

The authors declare there are no competing interests.

Huseyin Tunc conceived and designed the experiments, performed the experiments, analyzed the data, prepared figures and/or tables, authored or reviewed drafts of the article, conceptualization; Data curation; Formal analysis; Investigation; Methodology; Resources; Software; Validation; Visualization; Writing - original draft; Writing - review & editing, and approved the final draft.

Murat Sari conceived and designed the experiments, authored or reviewed drafts of the article, writing - review & editing, and approved the final draft.

Seyfullah Kotil conceived and designed the experiments, prepared figures and/or tables, authored or reviewed drafts of the article, conceptualization; Methodology; Visualization; Supervision, Writing - review & editing, and approved the final draft.

The following information was supplied regarding data availability:

All data and necessary codes are available at Github and Zenodo: https://github.com/tnchsyn/multistrainhivmodel; tnchsyn. (2023). tnchsyn/multistrainhivmodel: v1.0 (v1.0). Zenodo. https://doi.org/10.5281/zenodo.7547299.

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
