# Peer review of "Machine learning aided multiscale modelling of the HIV-1 infection in the presence of NRTI therapy"

_PeerJ, doi:10.7717/peerj.15033_

## Round 0.1 · original submission · Major Revisions

We have received comments from 3 reviewers. The first reviewer has very critical remarks. Please take into account all the suggestions, especially on the references and the figures update.

Reviewer 1 ·

Basic reporting

Summary. Because HIV-infected patients typically carry viral variants with many mutations, predicting impact of a given treatment (that typically consists of several drugs) may be difficult. Stanford database contains information on sensitivity of HIV strains to nucleotide reserver transcriptase inhibitors (NRTIs). The authors mined the database to predict impact of combination of several HIV mutations on overall susceptibility of variants to these NRTIs. These predicted "profiles" were then used in a mathematical model of HIV dynamics to predict impact of antiretroviral treatment with one or two drugs on viral control and progression to AIDS. The authors reaches somewhat expected conclusion that starting treatment late or with poor compliance will result in faster progression to AIDS.

Experimental design

None

Validity of the findings

Seems ok but see specific comments.

Additional comments

Main comments

1. While academically it is interesting to consider treatment with only NRTIs, it is not standard clinical practice. Therefore, the whole premise of the paper is outdated. Citing from the WHO website " tenofovir disoproxil fumarate (TDF) + lamivudine (3TC) (or emtricitabine, FTC) + efavirenz (EFV) 600 mg as the preferred first- line antiretroviral therapy (ART) regimen for adults and adolescents. Since that time, scientific evidence and programmatic experience have accumulated on the use of dolutegravir (DTG) in both first- and second-line ART, including during pregnancy and tuberculosis co-treatment, and for children. In 2018, these guidelines were reviewed to provide updated guidance on preferred option for these populations which now include DTG and raltegravir (RAL), and updated recommendations on using ARV drugs for HIV post-exposure prophylaxis. Annex 3 above links to the ARV dosing guidance for children, adolescents and adults."

https://www.who.int/publications/i/item/WHO-CDS-HIV-18.51

Another point that is also relatively "outdated" is focus on progression to AIDS. Typically, if viral load is not suppressed (i.e., not below the detection limit), patients will progress to AIDS (and treatment is considered as a failure); it does not matter how long it would take. So, full suppression of viremia is the goal - which seems to be unachieved in the simulations (e.g., Fig 5).


2. The reporting of how well the ANN models predict the outcome should not be judged by MSE or R. There are better metrics as such ROC curves (https://en.wikipedia.org/wiki/Receiver_operating_characteristic). The data plotted in Fig 1 are unclear - what are these? Are these IC50 values? x and y axis scales are different which makes it difficult to compare. Caption is poor -- too short and not very informative. It needs to include more detailed information.


3. The model seems to consider virus dynamics in 1 ul of blood. This is incorrect. The dynamics occurs in the whole body, where mutations will be much more frequent. In addition, expanding the model just to blood is not fully correct since most of infected cells are located in tissues, and thus, mutants are likely to be generated in lymph nodes (PMID: 10358770). Furthermore, the model ignores latency (latent infection of target cells and reactivation). Latency is thought to be the major problem requiring life-long treatment. It was not clear why this important aspect of HIV dynamics during treatment was ignored.


4. The difference between models 1 & 2 in "general simulation results" is not very well justified. It is obvious that during treatment start, there will be several strains in the patient. Why are we interested in the model 1? Also, the statement that "Figures 8-9 show excellent accuracy" (line 288) should be supported by a more rigorous analysis, e.g., based on ROC curves or similar.


5. Predicting resistance profile of a strain given some mutations is very interesting. But this needs to be compared to clinical practice (perhaps) and definitely to previous studies addressing similar problem (e.g., search on pubmed with "predicting hiv resistance from sequence" gave 67 hits including PMID: 34241771, 27992346, 29671808, 12499299). My understanding is that when sequencing of HIV is available, physicians do determine if specific mutations are present, so then treatment can be tailored to an individual. How much better would be the machine learning-based analysis you performed? Perhaps that has to be the benchmark. But also compare to the wealth of other approaches addressing similar problem (cited above).


Minor comments


It is unclear if authors consider epistasis in mutations. Some work showed that there epistasis in HIV mutations (PMID: 15567861).

Virus mutation rate was considered to be constant. It that a good assumption given specific mutations authors consider? Some of these may be more or less likely. This needs to be better addressed.

Table 4 lists potential impact of the drug combinations for treatment. What about their costs and side-effects? These may be important to consider which treatment is appropriate.

The overall conclusion is that starting treatment late or having poor compliance results in faster AIDS phase. Isn't that obvious? How is the model useful here?

Graphs/Figures need improvement. For example, Figure 6 - I cannot read anything in it. Figure 5 could be changed with x scale being in years, not days. Overall, scales must in the total body estimates, not per ul. Also, captions for all figures must be more detailed explaining what was done, how, referring to equations when needed, etc.


I don't think that "unique fitness cost of each mutant strain" is the inevitable criticism (line 333). I think it may be an ok assumption. But there are other criticisms - see above.

·

Basic reporting

The paper is very clear and requires only minor grammatical errors, the language, literature used is very crisp and appropriate.

Experimental design

- This paper is a original contribution
- Questions/abstract is very well defined, relevant and meaningful.
- Rigorous investigation has been performed to a high technical & ethical standard.

Validity of the findings

- Experiments and the results are very accurate.

Additional comments

Only minor literature editing is needed.

Reviewer 3 ·

Basic reporting

The manuscript is well written with clear description, appropriate literature references, and sufficient background. The authors also made their analysis code publicly available via GitHub. This is very important for transparency and reproducibility, so really happy to see the authors sharing their code.

The manuscript could be further improved with following aspects:
1) Line 166: "Figure 3 and 3"?
2) Line 223: "In 4"?
3) Line 265-267: Comparison between Figs 3 and 6 is not intuitive for readers. The authors could list specific examples and highlighting sub-panels of the figures to better illustrate their point.
4) Figure 6: legends and axis are too small to read.

Experimental design

The authors did a good job analyzing and modeling the data to address the well-defined research question. The experimental design and method description could be improved in the following aspects:
1) How did the authors select hyperparameters for ANN? The model performance presented in the paper should be done on a hold-out dataset which has not been used for parameter tuning and model selection.
2) In section "An artificial neurla network model for isolate-fold change relation", the authors should first show linear regression does not achieve good performance before using ANN. Otherwise, it does not justify the use of a much more complicated and black-box model when a much simpler linear relationship is sufficient for modelling.
2) In section "General Simulation Results", given the number of input features, use of ANN seems to be a way overkill. Using linear regression can also help infer the importance of each feature.

Validity of the findings

The authors did a good job providing underlying data. Conclusions from this manuscript are well stated and limitations of current study are well presented.

---

## Round 0.2 · Major Revisions

Thanks for the manuscript update. Unfortunately the comments by reviewer #1 were not taken into account. The work should be revised.

Reviewer 1 ·

Basic reporting

N/A

Experimental design

N/A

Validity of the findings

N/A

Additional comments

Most of my comments on the previous version of the paper were not fully addressed.

1. Treatment with only NNRT inhibitors is not the current way of treating patients.

2. The model does not include dynamics of the virus in tissues. We know that tissues (e.g., lymphoid tissues such as lymph nodes and spleen) harbor the most of the virus in the body (PMID: 10358770).

3. Basic conclusions of the modeling are trivial - e.g., failure occurs if patients are not compliant.

4. Drug treatment should result in complete viral suppression. Otherwise, this is a failed treatment and should not be continued. Modeling this is irrelevant to the current practice.

5. Latency is a key at causing viral rebound after stopping the treatment. The statements about this in the text (lines 152-158) are contradictory - one line states that T cell latency is not important and another states the opposite. Also, there is no strong evidence that macrophages carry most of the proviruses. Currently, proviral DNA is measured in T cells.

Reviewer 3 ·

Basic reporting

The authors have well addressed my comments.

Experimental design

The authors have well addressed my comments.

Validity of the findings

The authors have well addressed my comments.

Additional comments

The authors have well addressed my comments.

---

## Round 0.3 · Minor Revisions

Thanks for the manuscript update and detailed answer. The reviewers have no more critical remarks.

However I add my editorial comment on the presentation style and references:

See lines 50-51: bulk citation : “(Hadjiandreou et al., 2007; Perelson and Nelson, 1999; Dixit and Perelson, 2004; Rong et al., 2007; Sutimin et al., 2017; Wu and Zhao, 2020; Chen et al., 2021)”
Please cite these papers separately, add detail, which work consider which model type.
Cite older publications (before 2007) first, then describe new approaches to the modeling (by 2017, 2021).
Add more recent reference.

Line 43: Tarasova et al., 2018 - may cite in addition more recent work of the same group on same topic -
See
https://pubmed.ncbi.nlm.nih.gov/36675202/
https://pubmed.ncbi.nlm.nih.gov/36674980/
Machine learning is fast developing field, it is worthy use recent work on HIV models in this field.

Line 61: “Stanford HIV drug resistance database...” - add in-text reference to this database.

Line 97 and below: the lines in the paragraph are not numbered.
Add reference or web-link to “MATLAB program”. Note release number,
Formula Fold Change = ANN X (isolate)
Looks non-mathematical - add some formalism (interval of values, input data type (binary vectors?)

Line 123: “various studies in the literature” - it is also kind of bulk citation.
What differ in these models?
(Dixit and Perelson, 2004) is oldest work from the cited. The reference is repeated several times in same paragraph.

Use ‘ANN’ in Italic font (paragraph above line 132)

Table 2 has repeated reference in whole last column (Hernandez-Vargas and Middleton (2013) )
Need not repeat, just make table note. Indicate parameter variation instead (it is not given for all rows)

Line 371 and 381 - the figures cited together “Figures 8-9”. Please write separately, what is visible in Figure 8 and what is Figure 9.

Line 415: “REGA (Van Laethem et al., 2002), and ANRS...” give abbreviation REGA, ANRS in full. Add details. It is not clear from the title what kind of data it contains.

I believe these minimal comments could be addressed soon.

Waiting for the reply to the comments and resubmission of the revised manuscript.

Best regards
Prof. Yuriy Orlov
Academic editor

Reviewer 3 ·

Basic reporting

The authors have well addressed my comments.

Experimental design

The authors have well addressed my comments.

Validity of the findings

The authors have well addressed my comments.

Additional comments

The authors have well addressed my comments.

---

## Round 0.4 · accepted · Accept

Thanks for the updates. The manuscript had only technical remarks at the previous stage. All the remarks were considered. The manuscript should be accepted for publication in its current form.